# Strange attractor of a narwhal (*Monodon monoceros*)

**Evgeny A. Podolskiy**[1]*, **Mads Peter Heide-Jørgensen**[2]

**1** Arctic Research Center, Hokkaido University, Sapporo, Japan, **2** Greenland Institute of Natural Resources, Copenhagen, Denmark

* evgeniy.podolskiy@gmail.com

**Data Availability Statement:** All narwhal depth measurements are freely available online [https://doi.org/10.5281/zenodo.6522945]. Sea-ice concentration was retrieved from NOAA scientific data depository (AVHRR Pathfinder Version 5.3

## Abstract

Detecting structures within the continuous diving behavior of marine animals is challenging, and no universal framework is available. We captured such diverse structures using chaos theory. By applying time-delay embedding to exceptionally long dive records (83 d) from the narwhal, we reconstructed the state-space portrait. Using measures of chaos, we detected a diurnal pattern and its seasonal modulation, classified data, and found how sea-ice appearance shifts time budgets. There is more near-surface rest but deeper dives at solar noon, and more intense diving during twilight and at night but to shallower depths (likely following squid); sea-ice appearance reduces rest. The introduced geometrical approach is simple to implement and potentially helpful for mapping and labeling long-term behavioral data, identifying differences between individual animals and species, and detecting perturbations.

## Author summary

While animal-borne ocean sensors continue to advance and collect more data, there is a lack of an adequate framework to analyze records of irregular behavior. For example, in the Arctic—there sea-ice is declining but is fundamental for the life cycle of many endemic animals—near-surface dive records are usually ignored, and continuous data are reduced to a maximum depth or similar. Here, we propose to transform our way of thinking about animal motion underwater by turning to a chaos approach and using a flowing geometrical shape to understand the full diversity of behaviors on an example of a satellite-tagged narwhal. Our method may help to assess the susceptibility of narwhal and other animals to sea-ice loss and climate warming.

## Introduction

The endemic to Arctic narwhal has an anomalously strong cardiovascular response to stress and extreme auditory sensitivity, and it is one of the most endangered Arctic species due to climate change, anthropogenic activities, and invasive species such as killer whales [1–5]. While famous for dives deeper than 1800 m, the narwhal life cycle, as for most other Arctic animals,

243 L3-Collated, Global, 0.0417∘, 1981-present, Daytime, 1 Day Composite) freely available at [https://coastwatch.pfeg.noaa.gov/erddap/griddap/nceiPH53sstd1day.graph]. The codes used for our analysis are available at [https://github.com/Jehiko/narwhal/].

**Funding:** This work was funded by the Greenland Institute of Natural Resources (MPHJ; https://natur.gl/?lang=en), the Danish Cooperation for the Environment in the Arctic grant 2013_01_0289 (MPHJ; https://um.dk/en/foreign-policy/the-arctic), the Carlsberg Foundation grant CF14-0169 (MPHJ; https://www.carlsbergfondet.dk/en), the Hokkaido University Support Program for Frontier Research (EAP; https://wwwchem.sci.hokudai.ac.jp/~active-matter/index_en.html), and the Arctic Challenge for Sustainability II, MEXT, grant JPMXD142031886 (EAP; https://www.nipr.ac.jp/arcs2/e/). The funders had no role in study design, data collection and analysis, decision to publish, or preparation of the manuscript.

**Competing interests:** The authors have declared that no competing interests exist.

is fundamentally linked to sea-ice, which is now disappearing catastrophically [6, 7]. Analysis methods ignoring the near-surface activity and focusing exclusively on discrete dive units are therefore inadequate.

In the analysis of dynamic systems, it is common practice to graphically reconstruct their phase or state space. This geometrical means of problem formulation, proposed by Poincaré [8], visualized by Lorentz [9], and applied to biology by Winfree and May [10, 11], created a new language and revolutionized multiple disciplines dealing with a wide spectrum of nonlinear, chaotic, irregular, and quasi-periodic processes such as turbulence, heart and brain oscillations, population-, immune-, and astrophysical dynamics, and market cycles [12–15]. In particular, in 1963, Lorentz discovered that deterministic equations representing convection in the atmosphere exhibit aperiodic and irregular oscillations, which never repeated exactly, depended sensitively on the initial conditions, and thereby were unpredictable in a long-term [9]. Such behaviour is known as *chaos*. Lorentz also discovered that chaos had structure. He showed that, in phase space, a wonderful butterfly-shaped structure emerged to which all neighboring trajectories converged. Such set of points is called an *attractor* [15]. Currently, chaos presents a challenge for modern machine-learning and statistical techniques [16–18].

When data are limited, as is usually the case, *time-delay embedding* can be used to reconstruct multi-dimensional dynamics of non-linear systems from a one-dimensional (1D) time-series [19, 20]. The key idea behind this method is that by sampling only one observable of a dynamical system (e.g., $x$), one can obtain "a faithful phase–space representation of the dynamics in the original $x$, $y$, $z$ space" [19]. Time-delay embedding reconstructs a phase–space picture from a time series using the value of the observable with its values at delayed times (e.g., $x(t)$, $x(t + \tau)$, $x(t + 2\tau)$; where $\tau$ is the lag time). For three-dimensional (3D) chaotic dynamical systems of Rössler and Lorenz, this method yields sets of points which are diffeomorphically equivalent to the originals. We hypothesise that for whale depth readings, such iterated mapping allows reconstruction of the three-dimensional subsurface movements of a whale, which may abstractly resemble original trajectories (see Materials and methods for a step-by-step guidance). This can help in identifying recurrent dynamics and predominant states of diving behavior. However, in studies of animal behavior in general, and aquatic locomotion in particular, this potentially powerful tool has not yet been widely applied [21, 22].

Biological sequence analysis involving Hidden Markov Models (HMMs) [23] has been increasingly applied in analyzing the diving behavior of marine and other mammals [24–28]. Here, we took a different dynamic view of continuous and presumably chaotic animal behavior through the lens of continuous chaos theory, which apparently matches to the task. Specifically, we introduced time-delay embedding to characterize the diving behavior of a narwhal, detecting diurnal patterns and seasonal trends, classifying activity types, and estimating associated time budgets. Such outputs are essential for understanding animal behavior, demonstrating a possibility of coherent and principled analysis.

In brief, we used chaos theory to analyze exceptionally long narwhal tagging data, acquiring a wide spectrum of behavioral states. Details of our approach and data are described in the Materials and methods (also see Supporting Information's Figs A–G in S1 Appendix). The same data set was previously explored using HMMs [26]. This provided a unique opportunity for comparison of two fundamentally different paradigms, and to highlight our approach as being competitive and transparent, requiring only three steps to graphically reveal all possible states rather than methodologically demanding HMMs.

In this first attempt to compare such dissimilar conceptual toolkits, it is premature to identify the main difference. We conjecture that a time-delay embedding is a data-driven method suited as an exploratory tool. In contrast, HMMs correspond to a probabilistic modelling approach suited to model switches between states. Moreover, even if there is no one-to-one

correspondence between the states identified using both methods, we presume that our method might help in selecting the number of states of HMMs. Thus, we advocate for future experiments to confirm this potential advantage.

## Results

### Behavior state–space and recurrent trajectories

Time-delay embedding transformed 83 d of depth readings into a flowing geometrical object (i.e., a manifold) representing all possible diving behaviors of the narwhal (Fig 1A), with each point neighboring a dynamically similar epoch (Fig C in S1 Appendix). The unfolded behavior space displays a filamentary vortex in the first octant of Cartesian 3D space with several key features. The "core" is the center of gravity or locus where all trajectories converge toward the origin [0, 0, 0]. The core is surrounded by three layers, a nebula formed of inner orbits, a region formed by outer orbits, and a repelling gap between the two orbitals. Each orbital is unstable, so the attractor is strange and intransitive. Furthermore, each orbital apparently has a repeatedly visited location outside the core (on the [x(t), x(t+2$\tau$)] plane; see Materials and methods for details). The axes at the intersection of the planes correspond to transient states, including transitions between the main regions and individual dives that were not part of bouts, as exemplified below.

Close inspection of individual trajectories reveals several main stereotypical patterns, which form the principal architecture of the global attractor and comprise the key repertoire of narwhal diving behavior (Fig 1B and 1C), as follows:

1. Prolonged "rest" from significant dives; i.e., an attractor near the origin, which is also a locus of the state–space.

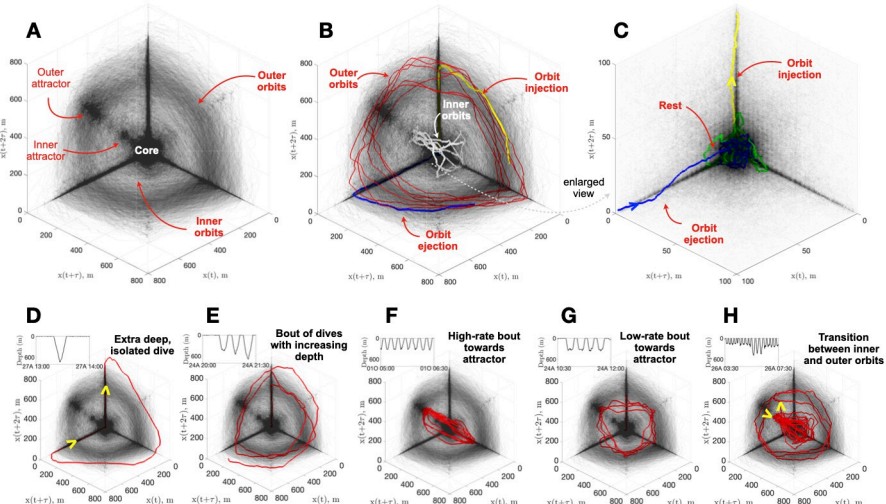

**Fig 1. Diving behavior state–space revealed by time-delay embedding. (A)** The locus of the vortex (the "core") is the main attractor of the system with inner and outer orbits having secondary attractors, with a gap between. **(B)** The most stereotypical trajectories, such as transitional injection into outer orbits (yellow), strongly oscillatory regions of outer orbits representing a deep-dive bout (red), transitional ejection back to the core (blue), and a secondary type of oscillatory region along inner orbits representing a shallow-dive bout (white). **(C)** Zoomed view of subpanel B, highlighting the "rest" state near the origin (green), and pathways toward outer orbits and back. Different examples of narwhal diving repertoire such as **(D)** isolated deep dive; **(E)** a telescopic spiral due to gradual change of maximum depth of each dive (outward or inward depending on the sign of change); **(F, G)** repeated visits to outer attractor with different rates; and **(H)** a transition between two orbitals (yellow arrows show inward and outward entries into a particular orbit).

2. Transition to dive session or bout; i.e., drop out from the rest attractor along the vertical axis x(t+2$\tau$) and injection into oscillatory orbits.

3. Dive bout; i.e., trajectories clearly detached from the rest attractor.

4. End of a bout; i.e., ejection from an orbital and return along the x(t) axis back to the rest attractor.

For convenience, we label these states, 1–4 above, as R, $T_{RB}$, B, and $T_{BR}$, respectively. Outer orbits, corresponding to deep diving bouts, are the most common type of dive in this sequence (Fig 1B). Inner orbits, corresponding to shallow diving bouts, are also possible (Fig 1B), so we distinguish the two types as $B_d$ and $B_s$, where the indices represent "deep" and "shallow", respectively. Transitions $T_{RB}$ and $T_{BR}$ make the corresponding corners along the intersection of the planes near the axes x(t) and x(t+2$\tau$) (Fig 1B) slightly denser than the x(t+$\tau$) axis, which is functionally different and visited for oxygen recovery during diving bouts. Visual inspection (Video A in S1 Appendix) reveals that, barring any transitions, there are three principal basins attracting the narwhal: R, $B_d$, and $B_s$. Later we show that the near-core region R has two distinctly different states.

Although the above outlines the global state–space portrait of diving behavior, there are other syllables and features of the narwhal diving repertoire. For example, there are isolated deep dives making characteristic butterfly-type loops in the state–space (Fig 1D), spiral-like trajectories of a dive bout with gradually changing depth (Fig 1E), bouts with high or low frequency where dive frequency and shape control the compression of orbits (from a semi-circle to an almost straight line; Fig 1F and 1G), a transition (or switch) from inner to outer orbits ($B_s \rightarrow B_d$; or $T_{BsBd}$) and back ($B_d \rightarrow B_s$; or $T_{BdBs}$) without passing the rest phase, R, and re-injection.

## Diurnal signals and attractor parameters

Above, we have described the phase portrait of all observed diving behavior states by mapping them as a 3D manifold (Fig 1 and Fig C in S1 Appendix). The question remains as how to use this graphical information to classify each time episode as a particular state. Automatic classification of behavior from time-series using a chaos-theory approach is a relatively recent development and has been used for human motion and fatigue detection, electromyogram signals, and accelerometer data of calves, *Drosophila*, and mice; however, empirical applications remain extremely limited [16, 21, 22, 29, 30]. Different features (or invariants) can be used to characterize reconstructed state–space; e.g., including correlation dimensions, approximate entropy, the Lyapunov exponent, Euclidean and Mahalanobis distances, and topological features [17, 21, 31, 32]. However, there are neither straightforward criteria to be used in selecting them [14], nor guidance in relation to diving behavior, so we departed from the most basic considerations (for detailed definitions of used chaos measures, see Materials and methods).

Binning measures of the dynamic attractor together with a dominant timescale in the system by hour of day reveals a clear diurnal variation (exemplified as violin plots in Fig 2). Specifically:

1. The kinetic energy, $E_k$, is related to the speed of the system along trajectories of the phase–space. This parameter can be viewed as a level of excitation for each particular state. Variation of $E_k$ suggests that diving activity is at a minimum around solar noon (between 12:00 and 15:00 h), while the narwhal is more excited during hours with less sunshine (Fig 2A). During twilight there are peaks in activity, particularly at 18:00–21:00 h. Overall, the day period (06:00–18:00 h) is characterized by a higher proportion of time spent in a low-energy

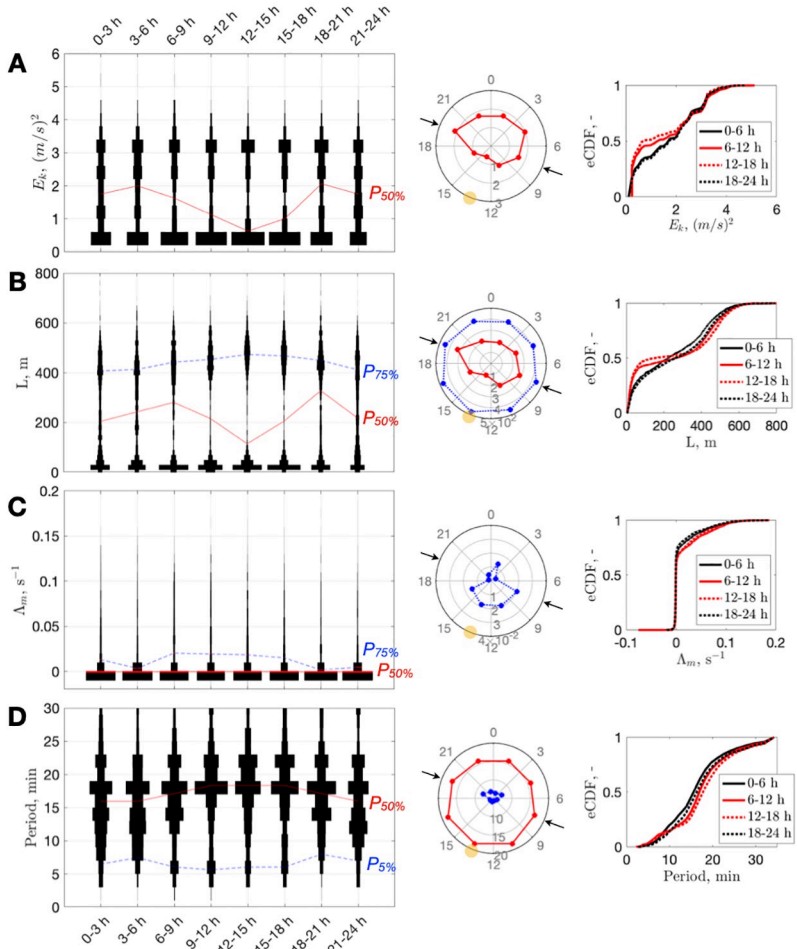

**Fig 2. Diurnal narwhal behavior.** Violin plots of **(A)** kinetic energy, $E_k$; **(B)** Euclidian distance, $L$; **(C)** maximum Lyapunov exponent, $\Lambda_m$; and **(D)** dominant timescale distributions by hour of day (using 3 h bins) with red and blue curves connecting percentile levels ($P_{50\%}$ and/or $P_{5,75\%}$). In the center column, the corresponding percentile levels are shown on a circular 24 h plot (representing the narwhal daily routine; yellow circle indicates the sun's culmination time, 13:29; arrows indicate the time of sunset and sunrise on September 25, 2013 the middle date of the dataset). Right side: empirical Cumulative Distribution Function (eCDF), for the same quantities computed using 6 h bins (black represents "night", red "day").

state (Fig 2A). Interestingly, energy partition changes from high at dusk to medium at night, then back to high at dawn (seen as thickening of the violin's neck by midnight in Fig 2A).

2. The Euclidean distance, $L$, is the absolute distance between the origin [0, 0, 0] and each individual trajectory point of the phase–space. It indicates that the narwhal is more likely to spend more time near the surface around solar noon than at night (Fig 2B). However, this is traded off for deeper dives in the afternoon compared with ∼70 m shallower dives at night.

3. The level of chaos in a dynamic system can be characterized by the maximal Lyapunov exponent, $\Lambda_m$ [33], which quantifies how fast trajectories separate (with plus corresponding to divergence toward chaos, minus to convergence, and zero to a stable limit cycle). $\Lambda_m$, indicates that irregular, chaotic behavior is more likely in daytime, with the likelihood of

less-chaotic behavior beeing higher at night (Fig 2C). Dimensionality, which represents self-similarity of a geometric object, has similar variability, increasing in the afternoon (S1 Fig).

4. Finally, the dominant timescale stays at around 17 mins, although departure toward longer periods is more likely in daytime and toward shorter periods at night (Fig 2D). Furthermore, daytime period lengthening is accompanied by the appearance of shorter periods (<10 mins). These features mirror diurnal variations in $E_k$ and $L$, and they are likely related to changing foraging conditions when the narwhal does not waste time at shallow depths.

A similar diurnal pattern of foraging has been observed for many low-latitude deep-diving odontocetes emitting more foraging buzzes at twilight [34]. However, for high latitudes (i.e., longer daylight), the existence of such patterns remains unclear. For example, no daily patterns have been found in blue whale calls around Antarctica, presumably due to a lack of light effects on krill concentrations [35], or in narwhal ingestion events in Scoresby Sound [36]. To our knowledge, only [37] found diurnal diving behavior of bowhead whales near Baffin Island (65˚N) at latitudes close to that of our study area (70˚N), with deeper dives in daytime in August, similar to our findings. Weddell seals at 66˚S also dive deeper in the afternoon, particularly toward the end of the austral summer [38]. Mid-water narwhal prey (such as squid *Gonatus sp.* and Polar cod *Boreogadus saida* [6, 36]) is known for diel vertical migration to deeper water in the afternoon [39, 40], possibly explaining the observed narwhal foraging behavior. In Scoresby Sound, squid is the most frequent item in narwhals' stomachs [36, 41].

Previous efforts to understand the same dataset using advanced HMMs [26] suggested the presence of diurnal patterns, particularly referring to transition probabilities from one state to another, such as: (1) a switch from deep to medium dives around midnight; (2) a backward switch around 06:00 h; and (3) a likely return to shallowest dives around noon [26]. These inferences were successfully recognized; i.e., the (1) and (2) correspond to thickening of the violin "neck" in $E_k$ and period shortening (Fig 2A and 2D), while (3) is a generally quiet condition around noon (Fig 2B). However, such discrete pieces of information are difficult to interpret biologically [26], and, presumably due to this, no comparisons with earlier literature or basic conclusions about the narwhal preference to rest longer but dive deeper at the solar noon were made. Our analysis thus provides a more comprehensive picture.

## Detection of states

The global co-variation of chaotic measures is shown in Fig 3A and 3B, which illustrates a convoluted continuity of states, with at least two dominant regions. The near-core region (≈<100 m) has the highest dispersion of increased dimensionality and divergence. At larger distances from the core, such dispersion disappears while energy increases; this region corresponds to strong oscillatory paths in which the narwhal repeats recurrent trajectories.

We acknowledge that chaotic-measures classification is a research question of its own [14, 16, 17], beyond the scope of this paper, although we aim to demonstrate the principal advantage of the approach. Here, for simplicity, we rely on Gaussian Mixed Models to perform unsupervised clustering of chaotic measures by iteratively increasing the number of components (Fig 3C). With more than four components, the Akaike Information Criterion no longer improves substantially (i.e., with likely overfitting), whereas with three components, the corner-twist of the parameter space near the core is merged into one and is thus insufficiently resolved (Fig 3A and 3B).

We also note that, as the transition from deep to shallower dive bouts (often around midnight) can be very smooth, it may seem natural to merge the two into one cluster (Fig E in S1

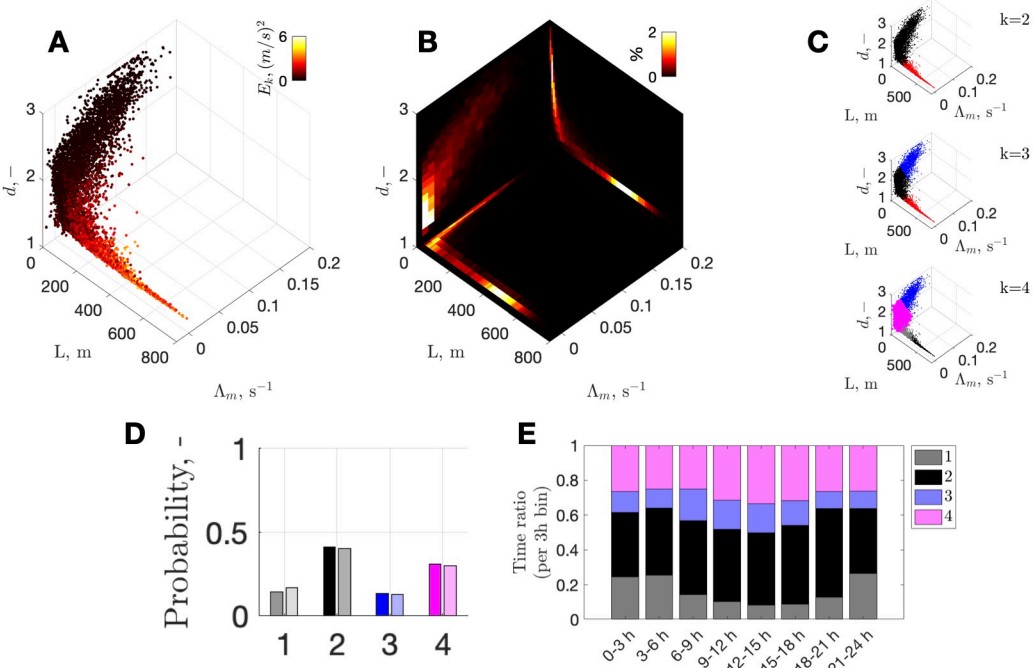

**Fig 3. From attractor properties to time-budgets and day partition. (A)** Overall parameter space of chaotic measures: Euclidian distances ($L$), maximum Lyapunov exponents ($\Lambda_m$), and attractor dimension ($d$) (color represents kinetic energy, $E_k$). **(B)** Example of corresponding density maps for each plane; colormap shows the number of points (%) within each bin (mesh: $dx$ 10 m by $dy$ 0.01 s$^{-1}$ by $dz$ 0.1). **(C)** De-clustered parameter space, using non-supervised classification based on Gaussian mixture models with 2, 3, and 4 components ($k$), revealing at least 2 distinct regions separated at $L$ ≈100 m. **(D)** Relative time budgets for each cluster. Points with the highest membership scores (>0.9) are shown in solid color (i.e., highest confidence); full clustering results (with possible ambiguity) are shown with faint color but are similar to high-score-only bins. **(E)** Partition of time between all clusters for each 3 h bin (full clustering results).

Appendix). However, we noted previously that the switch from shallow to deep ($B_s \rightarrow B_d$) can also be abrupt (Fig 1H). Therefore, distinguishing between the two types is justified. This outcome clarifies why the previous HMM-based study [26], which excluded <20 m depths from dive labeling, found four states unnecessary, with two or three equally acceptable.

It is noteworthy that there is no need to choose the number of states *a-priori*, relieving us of this notorious problem in HMMs [26–28], and allowing us to approach it after the system was characterized.

## Behavioral time budgets and labeling of time series

Selection of different clusters within the state–space domain (Fig 3C) allows retrieval of time-budgets and continuous labeling of initial data in the time domain. In this way we found that the narwhal was committed to diving 57% of time (clusters 1+2), spent the shortest amount of time, 13%, in the shallowest cluster 3, and engaged 30% of time in intermediate-depth activities of cluster 4 (Fig 3D), which were most prolonged during the longest hiatus in diving at the beginning of September, 2013.

Global partitioning of different activities by daytime (Fig 3E) indicates that the narwhal uses relatively more time for near-surface activities (clusters 3 and 4) in the afternoon, mainly at the expense of shallow dive bouts, $B_s$ (cluster 1). This is consistent with our earlier results indicating that daytime activity was less energetic and very shallow (median-wise). However, the partition changes with season: the diel signal is most pronounced in September (Fig F in

S1 Appendix) when there are ∼12 h of daylight, while near-surface time is reduced toward November. This is consistent with observations that time spent at 0–50 m depth is lower in winter, with conventional wisdom suggesting more intense foraging activity of narwhals in winter [6, 42]. It also indicates that the diurnal probability in narwhal behavior is time-varying and season-modulated.

Use of the above automatic clustering allows reproduction of results of equivalent quality to that of sophisticated HMMs; e.g., that two-thirds of time is spent in foraging dives was suggested previously [26]. Labeling is performed without arbitrarily chosen depth thresholds (e.g., to define any dive or "deep" dive), which are difficult to transfer from case to case as they might not represent behavioral preferences but rather local constraints due, for example, to bathymetry. Furthermore, continuous and full information is used rather than classification based on some preprocessed parameters such as maximum dive depth obtained after cutting off any animal activity at shallower depths. This is important when, for example, sea-ice constraints near-surface motion, which would otherwise be difficult to detect.

Closer examination of near-surface activities indicates that the narwhal rather abruptly reduced the number of shallowest activities (cluster 3) after 18:00 h, September 21, 2013 (Fig 4). Satellite-derived sea-ice concentration data on October 1, 2013 indicate the first appearance of sea-ice (with concentration ≥20%) in the middle of the Scoresby Sound fjord. Moreover, the median/mean depth of cluster-3 activities increased by approximately a body length, for ∼3–4 m (comparing one month before and 1 month after September 22, 2013). We interpret these two lines of evidence as the sea-ice pushing the narwhal toward less-shallow behavior (at least until migration toward the ice edge by the end of October, with less restrictive conditions

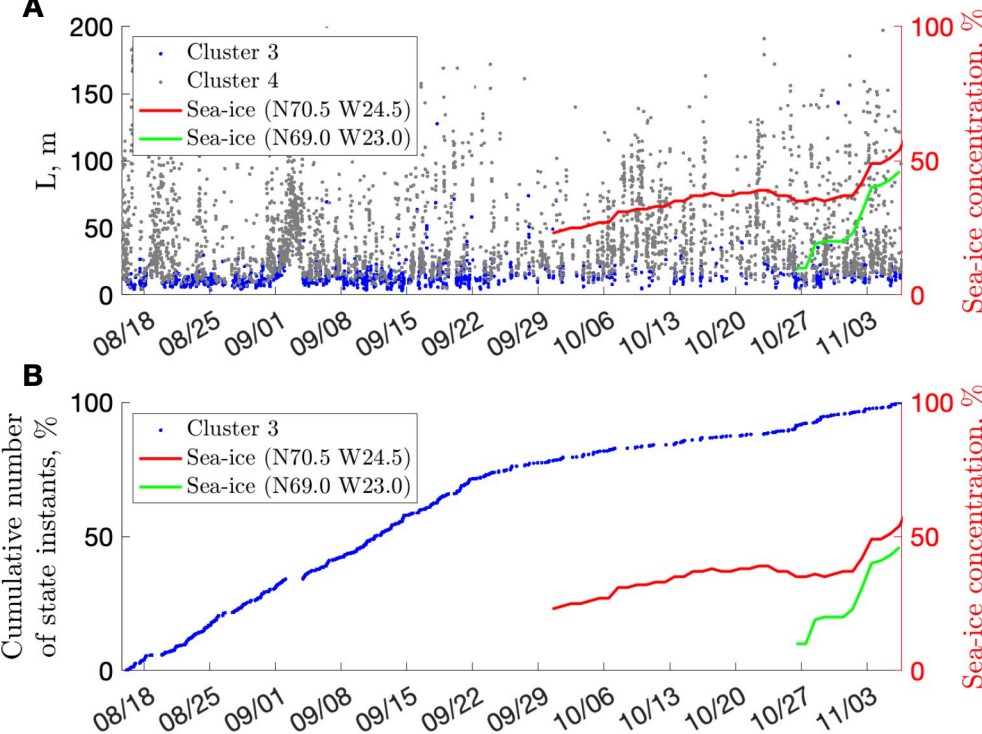

**Fig 4. Sea-ice and near-surface behavior. (A)** Variation of Euclidian distance for clusters 3 and 4 compared with sea-ice concentration (NOAA). **(B)** Cumulative number of cluster-3 instants (i.e., binary count such as 1 for "cluster present", 0 for "cluster not present"). Red indicates sea-ice concentration in the Scoresby Sound fjord (summering ground); green indicates the sea-ice concentration offshore (wintering ground), with typical grounds after [42].

than in the sealed branches of the fjord). This also means that we can determine the date of departure from the summering ground (e.g., September 15 was previously reported for Baffin Bay [6]).

## Discussion

Animal behavior analysis with chaos theory is useful but remains in its infancy. To our knowledge, there have been no previously published attractors corresponding to marine animal motion. While this work highlights the illuminating nature of such a perspective, it has focused on data analysis for a single animal, and thus needs to be extended for a larger number of individuals and species. This reflects a well-recognized limitation of chaos studies dedicated primarily to the celebrated Lorenz "butterfly" attractor [17]. Nevertheless, our data cover an unusually long period (records commonly cover no more than a few days; [26]), thus allowing detection of the time-varying nature of behavior on diel and seasonal scales.

We also note that a time-delay embedding might be useful for detecting an interdependence of multivariate time series, e.g., from multiple sensors or individuals [22, 43]. In fact, the theory of nonlinear dynamical systems offers many techniques for the diagnosis of synchronization and coupling [14].

Recognition of embedding as a powerful tool in studies of marine animals can be considered as the first step toward a systematization of similar animal-tagging campaigns, which are advancing rapidly with development of instrumentation [44]. This should inspire further studies of different animals. It is known that female narwhals perform more deeper dives [6], and mothers with calves may exhibit different behavior or there might be pod synchronization [24]. Considering the need to detect anomalies in the behavior of narwhals and other cetaceans in response to such perturbations as seismic air guns, ship traffic, climate change, and invasive species [4, 5, 26, 42], there is potential for different applications.

In practice, to achieve an unambiguous reconstruction of the state space, one should have data which document a recurrence of behaviour at sufficient sampling rates. (In theory, this is more complicated [45]). This implies that the suggested approach is also applicable to tracking records from avian and terrestrial species. There were activity-recognition studies on flies, mice, cows, and humans which support our claim [21, 22, 29].

Together with recent advances in machine learning toward improvement of classical embeddings by increasing dimensionality and retrieving governing equations from data, there are many unexplored opportunities [18, 46]. Extending this effort to all observed species may enable systematic and generalized investigations, as recently proposed for chaotic dynamic systems [17]. This may allow a dynamic view of life, not only in the ocean.

## Materials and methods

### Ethics statement

Narwhal tagging and related capturing was permitted by the Government of Greenland (Case ID 2010–035453, document number 429 926).

### Narwhal depth data

Data were from 83 d of depth readings (File A [50] and Fig A in S1 Appendix) at 1 s resolution from a narwhal tagged in 2013 at Scoresby Sound, East Greenland, via a satellite-linked time-depth recorder [26, 42, 47]. Mature male narwhal No. 3965 without a tusk (∼4.2 m long with mass of <1 t) was live-captured in a net, instrumented near the coast, and released within 30 min. 366 days after tagging, the animal was killed by local Inuit hunters as part of their

subsistence hunting. Details are provided in [36, 47]. The narwhal spent 50%, 70%, and 99% of its time within depths not exceeding 15, 106, and 556 m, respectively. During dives, the maximum depth was 911.5 m. The time spent at depths of around 400 m was slightly longer than at other depths.

To recognize the dominant timescales in diving behavior, we computed continuous 1D wavelet transfer (Fig A in S1 Appendix) using the Morse wavelet (Matlab default) [48, 49] (and references within). Results are shown as a movie together with embedding results (Video A in S1 Appendix). We also plotted the relationship between the instantaneous rate-of-depth-change and depth (Fig A in S1 Appendix).

## Embedding parameters

The state–space reconstruction of the underlying attractor was made via classical time-delay embedding [20, 51]:

$$y(i) = [x(i), x(i + \tau), \ldots, x(i + (m - 1)\tau)], \tag{1}$$

where $\tau$ is time-delay and $m$ is embedding dimension. Here we note that another possible set can be obtained by taking derivatives of a time series $(x, \dot{x}, \ddot{x})$. Such reconstruction is known as topologically similar to coordinates of Eq. 1 [19]. For activity recognition, instead of $x(t)$, some scientists also used accelerometer data [21, 29], which we did not collect.

To find appropriate embedding parameters we computed self-mutual information ($I$), auto-correlation ($R$), and false-negative neighbours ($FNN$) [45, 49, 52]. Furthermore, to reveal the time-varying variance of these estimates, we parsed the data into 6 h non-overlapping segments, computed the corresponding statistical features, and presented each result together with global median values (Fig B in S1 Appendix).

The median self-mutual information curve has its first local minimum at around 375 s (Fig Ba in S1 Appendix). It also has a main global peak at 1024 s, and a secondary smaller peak at around 510 s. The median autocorrelation curve drops below zero also at ~375 s (Fig Bb in S1 Appendix), but has only one main peak, at ~1050 s. Therefore, we chose a 375 s length as the optimal time delay, $\tau$, for embedding.

The free-diving whale has six degrees of freedom related to position (x, y, z) and orientation (pitch, roll, yaw). Narwhals are known to spin during foraging as known from multi sensor animal-born tag data on vocalizations and locomotion studied by [53]. However, we assumed that depth records, $z(t)$—usually the only data available to scholars—contain little information about orientation. Therefore, the number of dimensions necessary for embedding was $m = 3$. To verify this assumption, we computed $FNN$ using different values of $R_T$ [45] and found that with $R_T = 3$, the ratio of FNN becomes negligible in 3D (Fig Bc in S1 Appendix). For the higher values of $R_T$, FNN quickly flattens almost everywhere. Working with a planar view of data is indeed convenient, but our sensitivity tests confirmed that embedding in two dimensions is not satisfactory due to projection of data onto a low-dimensional space (folding), changing many points to false nearest neighbors.

Animated versions of the underlying attractor are shown in Videos A and B in S1 Appendix, with the former providing high-temporal resolution of raw data and reconstruction (2 min steps), and the latter giving a sense of long-term variability (6 h steps).

## Speed as epoch indicator

To emphasize the fundamental depth of reconstructed state–space and nonintuitive implications of each point position within it, we included the following considerations. To highlight that each point of a trajectory is likely to become a neighbor of a point separated in time but

from a similar dynamic epoch (e.g., descent or ascent), we colored points with the corresponding instantaneous diving speed ($\frac{dD}{dt}$). Speed is positive on descent, negative on ascent, and close to zero near the surface and at the lowest section of each dive (98% of absolute speed values are below 2.5 m s$^{-1}$; Fig A in S1 Appendix). This coloring procedure exposed a general clockwise flow with four regions of similar dynamic epochs (Fig Ca in S1 Appendix). Specifically, outside of the main rest attractor, during a diving bout, we observed the following patterns:

1. Episodes of short resurfacing involve the right plane [x(t+$\tau$), x(t+2$\tau$)] (white color, Fig Ca in S1 Appendix).

2. Rapid descents flow from the right plane toward the lower plane [x(t), x(t+$\tau$)] (red color, Fig Ca in S1 Appendix).

3. Points move and decelerate toward the third, left plane [x(t+2$\tau$), x(t+$\tau$)] (white color, Fig Ca in S1 Appendix).

4. Near the left plane, the trajectories switch to rapid ascent (blue color, Fig Ca in S1 Appendix), and return to the plane of brief "resurfacing".

Trajectories may pass through the corner between planes or "cut the corner" (i.e., early separation), moving in a ricochet fashion. To aid visual recognition of the global tendency of each epoch toward the main planes, we made the markers semi-transparent (Fig Cb in S1 Appendix). This revealed that most points plot on the three planes corresponding to three dominating epochs (Fig Cb in S1 Appendix). The fourth epoch of a "deep stay" is blurred into the neighboring epoch-planes owing to a short time at the depth (compared with a typically longer pause at the surface) or to varying dive depth.

In general, the described four epochs can be a part of different dynamic states, but outer orbits dominate the overall picture (Figs Ca and Cb in S1 Appendix).

## Dominant timescales

The dominant data timescale is centered around 17 min, as revealed by self-mutual information, autocorrelation, and continuous wavelet transform (Figs A and Ba, Bb in S1 Appendix), corresponding to quasi-periodic dives. We note that the characteristic time scale (from start-to-start) of a dive to 250 m can be nearly the same as to 450 m. The shorter period of 8.5 min (revealed by *I* in Fig Ba in S1 Appendix, but also visible in the animated continuous wavelet transform (CWT) results; Video A in S1 Appendix) corresponds to both the characteristic period of shallow dives (<100 m; such as around September 20, 22:49) and to a half-period of typical, deeper dives of 17 min.

The dominant period is a transcendental feature of both time and phase–space domains so, for consistency here, it is not attributed to chaotic invariants.

## Attractor properties

To characterize the underlying attractor, we computed several of its features, including Euclidian distance, $L$, kinetic energy, $E_k$, the maximum Lyapunov exponent, $\Lambda_m$, and the attractor dimension using a window of 30-min length (about two dominant periods), sliding in 10-min increments (Fig G in S1 Appendix). We quote these quantities not as actual estimates but as relative measures of system properties [14].

**Euclidean distance.** Using the analogy of atomic orbitals, the core (Fig 1A) can be seen as the most stable state with minimum potential energy, where the least kinetic energy is spent. Any displacement from this point of equilibrium corresponds to increasing energy and thus less stability. The inner orbits correspond to medium energy, and outer orbits to higher

energy. Trajectories departing further from the characteristic outer region correspond to the highest energy, least stable states. To quantify the magnitude of departure from the core, we computed Euclidean distance, $L$, between the origin [0, 0, 0] and each individual trajectory point:

$$L(t_i) = \sqrt{x(t_i)^2 + x(t_i + \tau)^2 + x(t_i + 2\tau)^2}. \tag{2}$$

**Kinetic energy along trajectories.** To quantify how energetic the motion along trajectories is, we computed the kinetic energy, $E_k$. Trajectory points are not equally spaced, so a state can change from $t_i$ to $t_{i+1}$ with different velocity, $v$, along each particular trajectory. Such velocity can be approximated as an increment of a 3D distance in the phase–space per unit of time, and then used to compute an equivalent of kinetic energy, $E_k = \frac{1}{2}v^2$. The corresponding visualization reveals the regions of high-energy trajectories, implying the fastest passage through those regions (Fig Cc in S1 Appendix). In particular, we observed that there is a concentration of high-energy trajectories along the plane passing approximately through the mid-axis $x(t + \tau)$ and the diagonal of the plane $[x(t), x(t + 2\tau)]$. A semi-transparent version of the same plot (Fig Cd in S1 Appendix) indicates that the neighborhood of the core, and pathways to it along the axes, have reduced energy. Note that the earlier-mentioned gap between the inner and outer orbitals (Fig 1B) can also be seen in Fig Cd in S1 Appendix.

In general, $E_k$ may be considered as the degree of the animal's engagement into diving. For example, at the onset of a new dive bout, the median $E_k$ level increases and at the end decreases; or during a dive bout, a single deeper dive leads to a jump in energy. The minimum $E_k$ is not zero; zero is expected for dead or trapped animals only.

Furthermore, time-series of such features as $E_k$ elevates our view to a higher hierarchical level involving longer timescales (Fig D in S1 Appendix). For example, rather than a characteristic period of 17 min, CWT shows timescales of between ∼2 and 27 h (Fig D in S1 Appendix), although, there is no typical timescale of intense prolonged diving (which is intermittent).

**Lyapunov exponent and attractor dimension.** Computation of $\Lambda_m$, for characterizing the rate of close trajectory separation, was performed according to [54]. Attractor dimensions were measured following [32]; the dimension, $d$, is a way to quantify self-similarity of a geometric object. $d$ is limited by the embedding dimension, $m$, which can be reached if the points are not confined to self-similar structure but smeared out over the state space like noise [14]. Other measures for characterizing chaotic systems exist, as well as approaches to computing basically "everything" [17, 21], however, we focused on the most basic measures, which are computationally less costly. For example, the approximate entropy (ApEn), measuring unpredictability or randomness of patterns in nonlinear time series [31], exhibited lower values during dive bouts and diurnal variation similar to $\Lambda_m$ and so has not been included here.

**Clustering and classification.** All computed properties ($E_k$, $L$, $\Lambda_m$, and $d$) were grouped through unsupervised learning based on Gaussian mixture models. This involved use of diagonal and shared covariance matrices, with random seed and fuzzy (soft) clustering, avoiding observations that could also be assigned to other clusters. Akaike's Information Criterion (AIC) was referred to for guidance and showed that more than four components for any type of model do not lead to obvious improvements.

One-to-one comparison of chaos-based classification versus HMM-based classification illustrates the quality of decoding (Fig G in S1 Appendix), and demonstrates how our approach organically includes any in-between episodes of re-surfacing as part of a particular dive bout, thus making dive and brief rest integral parts of the same dynamically coherent episode. However, questionable (subjective) identification is possible in both methods; e.g., very shallow and

isolated deep peaks can both be labeled as middle-depth peaks by HMMs, whereas chaos-based classification may attribute brief transitional time segments to inappropriate classes (Fig G in S1 Appendix). The latter stems from the unsupervised clustering procedure. Supervised classification may improve this when expert-labeled data sets or acoustic records are used for identifying borders within attractor properties. Finally, classification is generally achieved with a smaller number of variables than in HMMs ($m$, $\tau$, time-window, plus a few corresponding attractor properties vs. 26–112 variables in HMMs) and via significantly less laborious procedures, reducing to three steps before informed classification: (1) find the two key embedding parameters; (2) embed; and (3), quantify the results. Embedding runtime is also usually short (1 min on Intel Xeon E5 at 3.5 GHz) but invariant computations take similar time as for HMMs (up to 1.9 hrs on Intel Xeon E5 at 2.7 GHz [26]).

## Supporting information

**S1 Appendix. Illustrations and item legends (Figs A–G, Videos A—B, and File A).**
(PDF)

**S1 Fig. Diurnal narwhal behavior (dimensionality).** Violin plots of dimensionality, $d$, by hour of day (using 3 h bins) with red and blue curves connecting percentile levels ($P_{50\%}$ and/or $P_{5,75\%}$). In the center column, the corresponding percentile levels are shown on a circular 24 h plot (representing the narwhal daily routine; yellow circle indicates the sun's culmination time, 13:29; arrows indicate the time of sunset and sunrise on September 25, 2013 the middle date of the dataset). Right side: empirical Cumulative Distribution Function (eCDF), for the same quantities computed using 6 h bins (black represents "night", red "day").
(PDF)

## Acknowledgments

M.P.H.J. thanks his colleagues and Inuit hunters for support in the field; E.A.P. thanks Y. Mitani and M. Otsuki for literature advices.

## Author Contributions

**Conceptualization:** Evgeny A. Podolskiy.

**Data curation:** Mads Peter Heide-Jørgensen.

**Formal analysis:** Evgeny A. Podolskiy.

**Funding acquisition:** Evgeny A. Podolskiy, Mads Peter Heide-Jørgensen.

**Investigation:** Evgeny A. Podolskiy, Mads Peter Heide-Jørgensen.

**Methodology:** Evgeny A. Podolskiy.

**Project administration:** Mads Peter Heide-Jørgensen.

**Resources:** Mads Peter Heide-Jørgensen.

**Visualization:** Evgeny A. Podolskiy.

**Writing – original draft:** Evgeny A. Podolskiy.

**Writing – review & editing:** Mads Peter Heide-Jørgensen.

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
