## [Decision Letter · Decision Letter 0]

1 Jul 2022

Dear Dr. Podolskiy,

Thank you very much for submitting your manuscript "Strange attractor of a narwhal (Monodon monoceros)" for consideration at PLOS Computational Biology. As with all papers reviewed by the journal, your manuscript was reviewed by members of the editorial board and by several independent reviewers. The reviewers appreciated the attention to an important topic. Based on the reviews, we are likely to accept this manuscript for publication, providing that you modify the manuscript according to the review recommendations.

Sincerely,

Bard Ermentrout

Associate Editor

PLOS Computational Biology

Natalia Komarova

Deputy Editor

PLOS Computational Biology

[LINK]

Reviewer's Responses to Questions

**Comments to the Authors:**

Reviewer #1: Please see the attached report.

Reviewer #2: review is attached.

**Have the authors made all data and (if applicable) computational code underlying the findings in their manuscript fully available?**

Reviewer #1: Yes

Reviewer #2: Yes

PLOS authors have the option to publish the peer review history of their article (what does this mean?). If published, this will include your full peer review and any attached files.

Reviewer #1: No

Reviewer #2: No

Figure Files:

Data Requirements:

Reproducibility:

References:

---

## [Decision Letter · Decision Letter 1]

23 Jul 2022

Dear Dr. Podolskiy,

We are pleased to inform you that your manuscript 'Strange attractor of a narwhal (Monodon monoceros)' has been provisionally accepted for publication in PLOS Computational Biology.

Best regards,

Bard Ermentrout

Associate Editor

PLOS Computational Biology

Natalia Komarova

Deputy Editor

PLOS Computational Biology

Reviewer's Responses to Questions

**Comments to the Authors:**

Reviewer #1: All issues raised were satisfactorily addressed, I do not have any further comments.

Reviewer #2: I'm very happy with author responses to both reviewers, and highly recommend publication.

**Have the authors made all data and (if applicable) computational code underlying the findings in their manuscript fully available?**

Reviewer #1: Yes

Reviewer #2: Yes

PLOS authors have the option to publish the peer review history of their article (what does this mean?). If published, this will include your full peer review and any attached files.

Reviewer #1: No

Reviewer #2: No

---

## [Editor Report · Acceptance letter]

1 Sep 2022

PCOMPBIOL-D-22-00691R1 

Strange attractor of a narwhal (Monodon monoceros)

Dear Dr Podolskiy,

I am pleased to inform you that your manuscript has been formally accepted for publication in PLOS Computational Biology. Your manuscript is now with our production department and you will be notified of the publication date in due course.

With kind regards,

Livia Horvath
